# A political economy of the tobacco supply chain in an Eastern Mediterranean country: The case of Lebanon

**Ali Abboud[1]©, Ali Chalak [ID][2]©*, Joanne Haddad[3]©, Mariam Radwan[2]©**

1 Department of Economics, Faculty of Arts and Sciences, American University of Beirut, Beirut, Lebanon, 2 Department of Agriculture, Faculty of Agricultural and Food Sciences, American University of Beirut, Beirut, Lebanon, 3 European Center for Economics and Statistics (ECARES), Université Libre de Bruxelles, Brussels, Belgium

© These authors contributed equally to this work.
* ac22@aub.edu.lb

## Abstract

The literature on tobacco has traditionally focused on health effects, public policies for tobacco control, and smoker profiles. However, there is a notable gap in understanding the supply chains and industry practices within the tobacco market. This paper addresses this gap by examining the structure of the tobacco market in Lebanon. Using an exploratory qualitative research approach, this paper maps the tobacco supply chain in Lebanon and investigates the interactions among various stakeholders, including key policymakers, regulators, researchers, and industry experts, as well as their underlying interests. Lebanon is a compelling case study due to its high smoking prevalence, the presence of a state-owned tobacco monopoly (the Regie Libanaise de Tabacs et Tombacs (Regie)), and the ongoing financial crisis that has affected various sectors, including the tobacco industry. The findings reveal three key issues: a complex political economy centered around monopolization and conflicting interests, the absence of a clear national strategy on tobacco leading to ineffective policy formulation, and inefficient tobacco cultivation practices requiring reforms for sustainable agricultural development. To address these issues, it is proposed to foster a more competitive and revenue-efficient tobacco market through the dissolution of the Regie monopoly via horizontal and vertical integration. This includes adopting an excise-specific tax on domestically manufactured tobacco goods and optimizing sales taxes on locally traded items. Secondly, expanding antitrust laws to encompass the tobacco industry and introducing legislative measures for fees and taxes are recommended to create an enabling environment for competition and revenue generation. Thirdly, reforming tobacco cultivation practices requires abolishing the current cultivation licensing framework, offering financial compensation to existing license holders, and supporting farmers in transitioning to alternative crops.

**Data availability statement:** Data cannot be shared publicly because it includes self collected data that requires ethical consideration including obtaining an ethical approval from the American University of Beirut (AUB) Institutional Review Board (IRB) prior to recruitment and data collection. Ethics Committee (contact via IRB) for researchers who meet the criteria for access to confidential data. Encrypted or non raw data may be potentially shared with specific considerations. Data availability statement: This study analyzes data collected as part of qualitative research, which cannot be shared publicly due to ethical restrictions. We are bound by these restrictions from sharing a de-identified data set or excerpts from the transcripts, as doing so would violate the agreement to which the participants consented. The data contains potentially identifying and sensitive information, and the Institutional Review Board (IRB) imposes these ethical restrictions. The qualitative data includes self-collected information that requires ethical consideration, including obtaining ethical approval from the American University of Beirut (AUB) IRB prior to recruitment and data collection. Researchers who meet the criteria for access to confidential data can contact the third party: the Ethics Committee through the IRB at irb@aub.edu.lb; +961 01 350 000, Ext. 5445 (https://www.aub.edu.lb/irb/Pages/Aboutus.aspx#:~:text=The%20IRB%20is%20the%20committee,the%20location%20of%20the%20research). The consent forms that we used with our participants include the following statement: "Your information that is collected as part of this research will not be used or distributed for future research studies, even if all of your identifiers are removed."

**Funding:** The project is funded by the University of Illinois at Chicago's (UIC) Institute for Health Research and Policy to conduct economic research on tobacco in Lebanon. UIC is a partner of the Bloomberg Initiative to Reduce Tobacco Use. The views expressed in this paper cannot be attributed to, nor do they represent, the views of the American University Beirut, UIC, the Institute for Health Research and Policy, or Bloomberg Philanthropies. The funders had no role in study design, data collection and analysis, decision to publish, or preparation of the manuscript.

**Competing interests:** The authors have declared that no competing interests exist.

## Introduction

Every year, tobacco use leads to the deaths of more than 8 million people, among whom approximately 1.3 million are non-smokers who are exposed to second-hand smoking [1]. The health dangers linked to tobacco smoking are widely acknowledged, making it a major public health threat worldwide [1]. Despite its severe impacts, including the deaths of over 8 million people annually [1], its being a leading risk factor for diseases like cancer, cardiovascular issues, and respiratory illnesses [2], and its contribution to poverty by diverting spending away from essential needs towards tobacco purchases [1], tobacco use remains highly prevalent and continues to rise [3]. Moreover, the tobacco industry continues to be one of the most lucrative industries globally ([4,5]).

Growing concerns about the economic and health consequences of tobacco use have spurred many governments to enact tobacco control policies and strategies aimed at curbing its consumption. These measures encompass taxation, public awareness campaigns, the implementation of health warning labels, restrictions on smoking in public areas, bans on tobacco advertising and promotion, and the provision of smoking cessation programs. Among these, raising tobacco prices through increased taxes has been identified as the most effective policy approach for reducing tobacco smoking prevalence [6–12]. This is of particular importance in the Eastern Mediterranean Region (EMR), where tobacco products are priced the lowest and have the second-lowest average excise tax per pack compared to other World Health Organization regions [13].

As indicated by the World Health Organization (WHO) [1], taxes imposed on tobacco products represent the most cost-effective strategy for mitigating consumption and the associated economic and healthcare expenses. The primary mechanism through which taxes achieve this reduction is by raising tobacco prices. According to WHO estimates, a 10% rise in tobacco prices would lead to a roughly 4% decrease in demand in high-income countries and a 5% decline in low- to middle-income countries [1]. Consistent with these findings, Salti et al. (2015) [14] observes that a 10% increase in cigarette prices, specifically, would result in an average consumption reduction of 4–6% in high-income countries and 2–7% in low- and middle-income countries [14]. This holds particular significance in developing countries, where it is projected that tobacco will contribute to an estimated 10 million deaths annually by 2030 [15]. Tobacco taxation provides additional advantages, including revenue generation and addressing external costs associated with tobacco consumption, such as illnesses affecting non-smokers and the expenses incurred in treating these diseases [16].

Within the tobacco literature, there has been a substantial focus on analysing public policies associated with tobacco control, including evaluating their effectiveness and conducting cost-benefit analyses concerning smoking economics. However, far less attention has been dedicated to examining the political economy of tobacco supply chains and associated, tobacco industry practices, as important as such an understanding is to the design and implementation of effective taxation policies. Such research would map production practices, distribution networks, marketing strategies, and the influence of multinational tobacco companies on global, regional and national tobacco consumption patterns, and would be instrumental for identifying entry point for tobacco control policies, taxation included, not least in view the industry's capacity to exert influence on policy-making bodies against implementing such measures [17]. The core question that this paper explores is how does the prevalent market structure impact the influence and relative power of the different stakeholders in the tobacco market.

The purpose of this paper is to assess the structure of the tobacco supply chain in Lebanon. We rely on an exploratory qualitative research approach to map the supply chain of tobacco products in Lebanon and understand the interactions and relationship between different

actors. Our sample of interviewees include key stakeholders, policy makers and regulators as well as researchers and experts in the field. Understanding the political economy of tobacco in Lebanon is crucial for effective tobacco control strategies, including taxation. The adoption of a qualitative approach offers a unique perspective that was not inspected in earlier studies on the tobacco supply chain in Lebanon. First, it allows a detailed exploration of relationships of interests and power between different stakeholders (The monopole, farmers, private companies, regulators, and legislators). Second, given the limited availability and reliability of quantitative data on tobacco in Lebanon, and the deliberate ambiguity in public reports, the collected qualitative data offers crucial amount of information that would otherwise be unavailable. As these issues are not unique to the tobacco sector in Lebanon, the methodological approach adopted in this paper offers a blueprint to analysing political economy relationship in other context with similar structural complexities and data limitations.

Studying Lebanon is pertinent for many reasons. First, due to the high prevalence of smoking. Based on World Bank indicators, 38.2% of individuals aged 15 and above in Lebanon currently use tobacco products on a daily or non-daily basis (The World Bank, 2020). This percentage exceeds the Middle East and North Africa (MENA) regional average by 19% and the average of low- and middle-income countries (LMIC) by 15.1% [18]. Considering non-permanent smokers and occasional experimenters with various tobacco products, the overall tobacco use rate would rise to 70% [19]. Second, Lebanon is one of the few countries that operate a state-owned tobacco monopoly (SOTM) known as the Regie Libanaise de Tabacs et Tombacs, commonly referred to as the Regie. The Regie's primary purpose is to generate revenue for the government through the manufacturing and sale of tobacco products. However, it shares many key features with Monopoly-Oriented Endgame Models (MOEM), which have been proposed as a potential approach to reduce smoking prevalence to below 5% within a specified timeframe [20]. In addition, we aim to further enhance our understanding of the tobacco industry's supply chain in Lebanon and the operations of the Regie. Lastly, the tobacco industry remains one of the most important sectors operating in Lebanon and continues to generate profits despite Lebanon's financial crisis.

Scholarly analysis of the tobacco industry in the Middle East remains limited, with no previous research focusing on the political economy of tobacco in Lebanon. However, some relevant papers shed light on this topic. Nakkash & Lee's 2008 [21] study examines how British American Tobacco (BAT) and other transnational tobacco companies (TTCs) accessed the Lebanese market through smuggling and legal channels amidst political instability, emphasizing the need for international cooperation to address cigarette smuggling in Lebanon and the Middle East. Additionally, Chalak et al.'s 2023 [22] landscape report on tobacco consumption and taxation in Lebanon provides background information on Lebanon's tobacco history and the supply and demand of tobacco products. Alaouie et al.'s 2022 [20] paper reviews tobacco endgame proposals, focusing on the Monopoly-Oriented Endgame Model (MOEM) and using the Regie in Lebanon as a case study to assess its alignment with MOEM features, highlighting key themes in governance and operational remit, and emphasizing the need for appropriate governance structures and financial incentives to suppress the expansion of the tobacco market. This study complements and add to this body of literature, by highlighting two significant issues: 1) the misalignment of policy objective between the monopole, with a profit maximizing objective, and the public policy makers who target several policy objectives, including maximizing of net public revenues, improvement of public health and environmental objectives; 2) the co-dependency between monopole and subsistence farming, by subsidizing and encouraging an inefficient system of substance farming, the monopole attain legitimacy as a social provider. Overall, this study offers a comprehensive picture of the

complex relationships in the tobacco supply chain in Lebanon, which should serve as a chart for any feasible and efficient policymaking.

Our study aims to fill this gap by collecting innovative exploratory qualitative data. This data allows us to comprehensively map the tobacco supply chain in Lebanon, enhancing our understanding of the interactions and relationships among various stakeholders as well as their underlying interests. We aim to provide valuable insights that can guide future policy decisions and contribute to a nuanced understanding of the tobacco landscape in Lebanon. Our research offers a robust framework for ongoing and future policy discussions, particularly regarding tobacco control measures. Given Lebanon's significance in this context, the recommendations derived from our findings may also be relevant to other settings and can be generalized.

This study employs an inductive approach which is motivated by the exploratory nature of the investigation into the political economy of the tobacco supply chain in Lebanon. Inductive research, as undertaken here, begins with specific observations and aims to lead to broader policy and conceptual insights. This approach is particularly suitable for understanding the complex but theoretically under-explored context like the tobacco industry and supply chain in Lebanon, where empirical data is scarce, and the relationships among stakeholders are complex and yet poorly understood. This study therefore attempts to propose a nuanced understanding of the interplay between state-owned monopolies, subsistence farming, and market dynamics within a politically sensitive industry.

The rest of the paper is organized as follows: Section 2 presents a concise background on the tobacco industry in Lebanon. Following that, we detail the methodology and sample employed for conducting a qualitative analysis of the Lebanese tobacco supply chain in Section 3. Section 4 presents our findings providing an overview of the tobacco industry's landscape in Lebanon. Subsequently, we delve into a discussion and present policy recommendations based on our findings. Finally, we conclude in Section 6.

## Background

The structure of the tobacco supply chain in Lebanon is challenging to understand. As early as the 1930s, Tobacco transnational Corporations (TTCs) aimed for a significant presence in the Lebanese market when a French entity acquired the Ottoman Regie's assets, forming the Compagnie Libano-Syrienne des Tabacs [21]. After the end of French rule in 1935, the control transitioned to the French company with Lebanese shareholders. This setup persisted until the 1950s, when the government reclaimed the Regie Libanaise des Tabacs et Tombacs (referred to as the Regie or RLTT), placing it under state ownership and the Ministry of Finance's oversight in 1959 [21]. Although initially not designed to be permanent, discussions of a planned tender emerged in 1961, and periodic assessments of the Regie's monopoly status ensued. Political instability and sectarian divisions within the country impeded resolutions. The legal standing of the Regie has remained uncertain up to the present day, with a significant challenge being the interconnection of the tobacco industry with influential groups that historically exchange favors with farmers and Regie employees for political support [21].

With respect to the tobacco supply, currently, approximately 80,000 dunams of land are dedicated to cultivating tobacco leaves, representing about 3.5% of the total arable land area. This cultivation yields an annual output of around 8,000 metric tons, with approximately 25,000 farmers across 458 villages involved in its production. Notably, 37% of these farmers are concentrated in the southern region of the country. However, alternative sources suggest a lower figure of 11,000 full-time tobacco farmers [22].

## Methodology and sample description

### Study design

This study uses qualitative research methods to explore key stakeholders' perspectives and experiences and understand the complexities and intricacies of the tobacco industry in Lebanon. Given the economic, social, cultural, and political factors influencing tobacco politics and supply chains, interviewing policymakers and researchers can reveal dynamics not captured by quantitative approaches. Additionally, qualitative findings can help provide insights into the motivations, challenges, and strategies of different key stakeholders in the tobacco industry which is crucial for planning targeted measures and nationally appropriate policies to regulate tobacco supply chains and reduce tobacco use [23].

According to Mathie and Camozzi (2005) [24], qualitative research is especially useful for "politically or socially sensitive topics", such as the dynamics of tobacco production and products smuggling. Furthermore, qualitative methods have been widely used in tobacco-related research. Newly published research delves into the governmental strategies and perspectives on tobacco control and its alternatives in Malawi. It involved semi-structured interviews with stakeholders working in the tobacco sector [25]. Moreover, recent studies aimed to explore challenges facing tobacco control policies in Indonesia and Australia by conducting interviews with national tobacco control experts including academics, community organisations, and government officials ([26,27]). Similarly, a study aiming to understand the dynamics of the waterpipe industry also utilized semi-structured interviews with representatives from various waterpipe companies [28]. In many studies, the use of qualitative methodology has been considered a strength given the richness of data that it provides regarding the tobacco industry especially when official documents that could shed light on the market dynamics are lacking.

### Population of interest & sample size

Qualitative data included interviews with 13 key informants in Lebanon. A list of participants was prepared and contacted by email. The interviews were conducted in person or virtually based on the participant's availability. The participants fall broadly into three categories: (1) Major stakeholders, (2) Regulators and policy makers, and (3) Researchers and policy advocates. The sample size was determined by considering the number of interviews needed to reach data saturation of key themes as well as feasibility given the timeline and resources available. At the beginning of the study, a minimum of 8–10 interviews were sought to meet study objectives or until data saturation was achieved. The final sample size depended upon data saturation.

Furthermore, the study incorporated a validation workshop that facilitated in-depth data discussions and elicited novel insights and feedback from a cohort of 10 participants, one of whom had previously participated in the interviews. This cohort encompassed representatives from ministries including finance, agriculture, and justice, as well as focal points in tobacco control, esteemed economics professors, and experienced tobacco researchers. This event not only served as an opportunity to engage stakeholders and experts, but also played a crucial role in enriching the analysis and drafting policy recommendations. It involved presenting the key findings obtained from key informant interviews, fostering open discussions, obtaining feedback, and addressing any uncertainties. The information gathered from this event was instrumental in refining the study's outcomes and shaping the subsequent policy recommendations.

### Recruitment and data collection

Potential stakeholders were identified through a desk review and literature research and were contacted ahead of time. Through purposive sampling, 16 national stakeholders who

had experience in tobacco market structure were selected for interviews. Purposive sampling was employed to select and recruit stakeholders based on their professional roles and direct involvement in this area, and to optimize the use of research resources. Only those who expressed interest in taking part in the semi-structured interviews proceeded to the interview phase. One invitee failed to respond to the invitation, and two declined participations, both of whom represented major tobacco companies. Interviews with stakeholders were led by the principal investigator (PI), Dr. A.C (Ph.D.), and the co-principal investigator (co-PI) of the project, Dr. A.A. (Ph.D.), and were transcribed verbatim in English and then analyzed. The PI and co-PI are expert researchers with a focus on tobacco, economics and applied choice analysis. The research assistant (M.R.) assisted in recruiting stakeholders, collecting data, and led the analysis of transcribed interviews. She also completed CITI training in Social and Behavioral Responsible Conduct of Research, a process facilitated by her prior training in qualitative data collection and thematic analysis during her graduate studies. A semi-structured approach was used for the interviews to explore individual perspectives of the tobacco market structure. Key informant interviews (KIIs) were conducted in person or through remote channels such as Zoom video meetings, depending on the respondents' preferences, and using unique and secure links for each individual. Interviews were conducted in private rooms in different facilities based on the preference of the stakeholder and lasted 60–90 minutes on average. Researchers took notes in the field following each interview which were utilized to aid in analysing the transcribed audio recordings. Data collection tools (interview guides) were developed by the research team members who are experts in the tobacco economics field, based on the study's objectives and considering existing literature. The interview guides included open-ended questions with probes to allow for free flow of discussion while exploring the key areas of interest for the study. The interview questions were pilot-tested, and minor modifications were made to ensure clarity. We have included the interview guide used in English in Appendix A.

The research team observed adequacy, credibility and reflexivity, which are the fundamental elements of trustworthiness in qualitative research. Regarding adequacy and dependability, the team conducted weekly discussions to ensure that data generation and analysis were appropriate for addressing the research questions. In terms of credibility, it was maintained through prolonged engagement with data, using transcriptions of audio-recorded interviews as a primary data source, and providing a summary of all main themes and sub-themes that emerged. In addition, to ensure researchers' reflexivity, data collectors were sensitive to their interaction with participants and applied thoughtful self-reflection before and after interviews with stakeholders. The study's reporting adhered to the guidelines outlined in the COnsolidated criteria for REporting Qualitative research (COREQ) checklist [29].

## Data management and analysis

Data analysis has been conducted concurrently with ongoing data collection and adopted an inductive coding approach. This approach was used to extract themes from raw data, enabling the identification of emerging topics relevant to the research question and the discovery of new, unexpected themes without any preconceptions or guiding hypotheses [30]. The analysis was conducted manually using a thematic approach described by Braun and Clarke (2006) [31]. Thematic analysis is "a method for identifying, analysing, and interpreting patterns of meaning ('themes') within qualitative data" (Clarke & Braun, 2017 [32], p.297–298). It has been used since it is flexible and can be adapted to our research question and sample size. It is also systematic, generates rigorous analysis, and provides interpretive depth [32]. It consists of six phases, starting from the immersion in the data, generating initial codes by systematically coding across the data set, identifying common themes and grouping codes into themes,

revising the themes and ensuring coherence, refining and naming the themes, until reaching the final step which includes embedding the themes in an analytic tale and writing the report. Findings and emergent themes were supported with quotes from individual interviews.

### Ethical considerations

An ethical approval was obtained from the American University of Beirut (AUB) Institutional Review Board (IRB) prior to recruitment and data collection. The recruitment of participants started on 8/8/2023 and ended on 6/10/2023. Confidentiality was maintained by using encrypted identification of individuals, limiting access to data to the interviewing team only, and storing data in secure locations. The purpose and procedures of the conducted interviews were explained to all study participants before seeking their consent to take part in the study. Verbal consent was obtained from all stakeholders, who also kept a copy of the consent form for their reference. Stakeholders were also asked to verbally consent, specifically for the recording and word-by-word transcription of the interviews. Participation of study participants was on a voluntary basis.

## Findings: Landscape of the tobacco industry in Lebanon

In the course of data analysis, five main themes emerged, shedding light on the dynamics and landscape of the tobacco industry in Lebanon as follows: The Regie: a bilateral monopoly, Tobacco farming: challenges and dynamics, big tobacco companies, distribution and wholesale, public oversight and public policy.

### The Regie: A bilateral monopoly

The tobacco market in Lebanon is fully controlled by the Regie. The Regie is a publicly owned state monopoly that is managed by an independent board, it operates under the supervision of the Ministry of Finance (MOF). In practice, however, there is only minimal interaction and active oversight, which results in the Regie being an independently administered monopoly operating with very limited oversight by any elected or governmental body. This assessment is based on statements from representatives of the Ministry of Finance (MOF) and the Regie. Several participants, mainly governmental representatives, highlighted the struggle in arranging and understanding the strange legal setting of the Regie:

> *"There had been some issues, but it was eventually settled that RLTT was managed as a private institution under guardianship of the MoF, provided that a Government Commissioner gets appointed by the MoF to monitor and supervise RLTT's work" (Governmental representative 1).*

The Regie has the exclusive right to provide licenses to farmers and wholesalers, to purchase locally produced tobacco leaves from Lebanese farmers, to purchase and import raw tobacco material, to regulate and set tobacco products standards, as well as to manufacture and distribute tobacco products. This was mentioned in the interview with the Regie representative when he introduced the Regie. However, it is common knowledge and is also available on the Regie's website. Based on the qualitative data generated from interviews with government officials, Regie representative, and researchers, it was emphasized that the transfer of Regie revenues to the treasury constituted some of the largest non-tax items of revenues in the yearly national budget:

> *"Back when the dollar rate was still at LBP 1,500, it (The Regie) used to generate 100 billion LBP in revenues for the treasury annually. This was equivalent to around 60 to 70 million dollars" (Governmental representative 1).*

Based on an interview extract with a researcher involved in tobacco monopolies and taxation policies, it is estimated that 5 to 6 billion dollars were transferred to the treasury between 1990 and 2020. These numbers align with the data provided by the Regie in its financial overview, which is reported in Fig 1. This statement has been further supported by the Regie representative who stated that:

> *"Since the committee was formed until today, we have given the state $7 billion net" (Regie representative).*

According to the "Citizen Budget" of 2022 which highlights the main fiscal figures and measures passed in the Lebanese budget, a booklet that was published in partnership with UNICEF, under the project "Enhancing Budget Transparency, Accountability and Inclusiveness in Lebanon", the Regie's profits amounted to 268.2 billion LBP in 2020 and 255.5 billion LBP in 2021, surpassing most total non-tax revenues except income from non-financial public enterprises and administrative fees, and almost equaling retirement deductibles.

The Regie's dominant position at the center of the tobacco industry in Lebanon allows it to generate large revenues. It is a bilateral monopoly, controlling the flow of products to downstream markets through its exclusive rights to produce and supply finished tobacco products. It also has significant market power on the upstream market, as the sole buyer of tobacco inputs (mostly tobacco leaves and other items used in tobacco products). According to official numbers published by the Regie, the total amount of revenues generated by the Regie between 1994 and 2021 is 4.177 billion USD, resulting in a net profit of 1.672 billion USD out of which 1.436 billion USD was transferred to the treasury. Fig 1 provides a diagram of the total sector revenues between 1994 and 2021 and their distribution across various components.

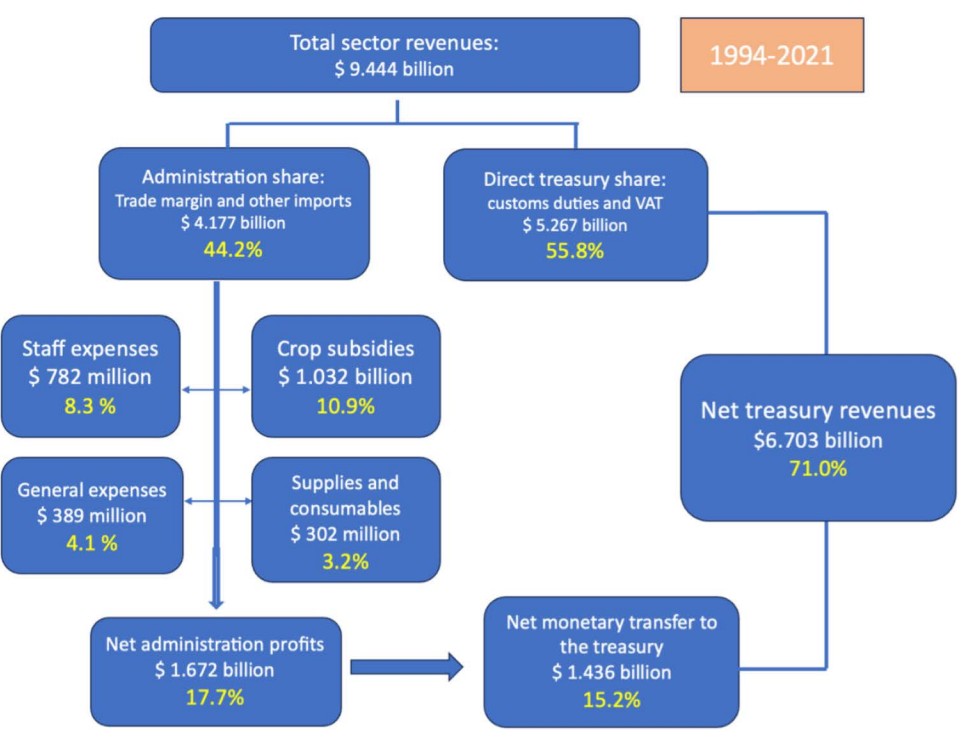

**Fig 1. Lebanese Regie financial overview.**

The behavior of the Regie is not fully monopolistic. More precisely, it does not fully take advantage of its dominant position on the upstream market to maximize profit; on the contrary, the Regie runs a subsidy program for tobacco farmers. The subsidy comes in the form of a negotiated price that is 60 to 100 percent higher than the global market price for tobacco leaves. The is inferred from interviews with both the Regie representative and the farmers union representative. Between 1994 and 2021, the subsidy cost was 1.032 billion USD, as reported in Fig 1.

In addition to the direct subsidy to farmers, the Regie runs an indirect subsidy and community support system, by financing the provision of various public goods and services, such as community centers, public libraries, digging and powering of water wells, scholarships for university students in Lebanon and abroad and other small projects satisfying needs of local communities. The main beneficiaries of these projects and initiatives are tobacco farmers and other residents of districts and villages with substantive tobacco farming:

> *"It (The Regie) implemented a rural development project. Every year, it implemented a project; in the South we had around 10 projects and maybe 10 in Northern Lebanon and Beqaa. It has implemented more than 100 projects over the years; these were either water collection ponds or agricultural roads or pumps to extract water from ponds. These projects were the backbone of agriculture in general not only tobacco farming" (Tobacco farmers syndicate representative).*

These programs have been subject to scrutiny by various stakeholders including parliamentarians, government career administrators and public policy advocates. The lack of transparency and regional inequality in spending of what many perceive as public funds outside of a national budget and without mandate from an elected body, is one of the main objections that are often brought up in the public debate and was reiterated by several interviewees. There is no clear nor publicly available account of the amounts spent by the Regie on these projects, nonetheless, under the tab of general spending, the Regie numbers show that they spent 389 million USD between 1994 and 2021.

Tobacco production is one of the major functions of the Regie. Their major brand—Cedars—had more than 50 percent of cigarette sales market share in 2019 [33].

In recent years, the Regie introduced another local cigarette brand, Byblos. More importantly, they negotiated and obtained the rights to produce international cigarette brands under license. In total, in its multiple facilities, the Regie has now thirteen lines of productions for the major international tobacco brands that sell in Lebanon. As such, Lebanon practically no longer imports cigarettes.

> *"Currently, the Regie has like 15 lines of production in Lebanon and they are proudly saying that most of the international brands are locally produced under the supervision of this big tobacco industry" (Researcher 1).*

> *"We managed to come up with an excellent formula. But we won't get into details concerning this (…) It is 100% transparent and everyone is happy. It first started with Imperial. Imperial was the first to manufacture here. Gauloises, Gitanes. And now honeyed tombac entered the scene. Many brands are manufactured here, like Fakher, Mazaya, and Nakhle" (Regie representative).*

Fig 2 presents an overview of the volume of tobacco cigarette imports and exports from 1961 to 2021, highlighting Lebanon's nearly negligible imports of cigarettes.

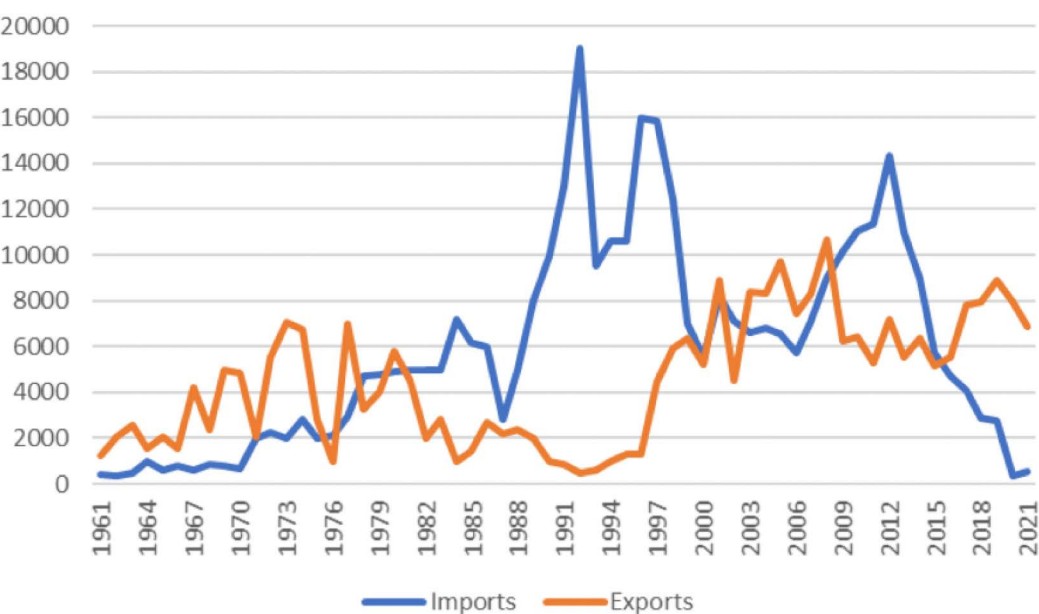

**Fig 2. Tobacco cigarettes imports and exports volume (Adapted from [ 22] based on FAO, 2022).**

The details and conditions of the licensing agreements between the Regie and the international tobacco companies are confidential. In our interview with the Regie representative, no details of the agreement were revealed, but the representative stated with confidence that the deal is favorable for the Regie and Lebanon. It is also not clear what are the implications of this agreement on tax status of these locally produced international cigarettes, especially that custom taxes used to constitute a significant amount of the taxes levied on these brands of tobacco.

## Tobacco farming: Challenges and dynamics

Tobacco farming has a long tradition in various areas of Lebanon. In the last five decades, it became concentrated in the South of Lebanon and the governorate of Akkar in the North of Lebanon. Tobacco cultivation in Lebanon started during the Ottoman era. The French mandate authorities (1918–1943) further encouraged tobacco farming and introduced several steps to organize the activity [34]. Most of these early regulations and organizational structures carried into the years of post-independence. In these earlier times, tobacco farming was subject to various forms of clientelist exploitation, but in the 1970s, a small farmers movement succeeded in weakening the control of large landowners over tobacco production. During the years of war and occupation of South Lebanon (1975–2000), tobacco took on a new meaning, becoming a "crop of resistance", providing a source of sustenance for the impoverished population amidst ongoing violence [34].

To engage in tobacco farming, farmers must obtain a cultivation permit from the Regie. As explained by Hamade (2014) [35], this permit remains active only if the farmer produces more than 200 kilograms per year for not less than 2 years in a row. The locally produced crop is then bought by the Regie. Only 5% of Lebanese raw tobacco is used in the local tobacco production, the remainder is exported:

> *"Lebanese tobacco leaf is not used in the local market, only 5% of it is found in the local market, given the fact that they changed the blend. Lebanese like the American blend"* (Researcher 1).

Fig 3 illustrates the import and export value of unmanufactured tobacco, confirming that the majority of domestically produced tobacco leaves are exported, with almost none being imported.

Nowadays, it is estimated that an area covering 80 km$^2$ is dedicated to cultivating tobacco leaves in Lebanon, constituting approximately 3.5% of the total cultivated land. As per the official data released by the Regie, the annual aggregate production amounts to approximately 8,000 metric tons, cultivated by roughly 25,000 farmers in 458 villages. A significant portion, 37%, is concentrated in South Lebanon, contributing to nearly 57% of the overall output [37].

The accuracy of the reported production numbers is questionable. The official numbers cited above likely reflect total potential production quantities based on Regie licenses to farmers. There was an agreement among the researchers interviewed that production has been steadily declining. The interviewed Regie representative confirmed the declining production trend as well, stating that in 2022, the total actual production was around 3,000 metric tons, and they expect it to be the same this year. The low returns and profitability of tobacco farming is the main reason for this decline in production, with many farmers quitting tobacco cultivation and switching to other types of crops or migrating out of rural areas to seek non-agricultural jobs.

There is a general agreement that tobacco farming is inefficient and that local farmers are unable to produce high quality tobacco at a competitive price compared to the international markets. The agricultural landscape itself poses difficulties, as acknowledged by the Regie representative, stating:

> "Our land is not agricultural… You must select the varieties you can work on. You can't compete with the tobacco agriculture; we are making it survive by subsidizing it" (Regie representative).

The subsidy system maintained by the Regie is the main lifeline for survival of tobacco farming in Lebanon. The tobacco farmers syndicate's representative acknowledged this, indicating that in their most recent round of negotiation with the Regie, they secured an average price of $5 per kilogram. However, the price per farmer could vary depending on the quality of their

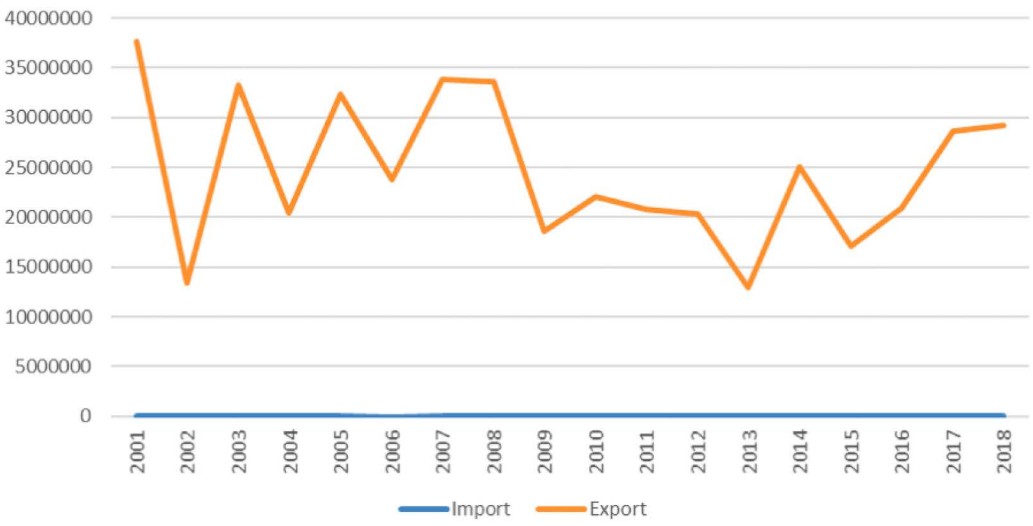

**Fig 3. Import and export value of unmanufactured tobacco (Adapted from [ 22] based on data from WITS, 2022 [36]).**

crop. This price was $2 higher than the global market average. The syndicate representative expressed their hope that the Regie will increase the subsidy in future years to make tobacco cultivation financially sustainable for farmers.

The inefficiency in tobacco cultivation spurs from its practice on small, fragmented plots. As expressed by the tobacco farmers' syndicate representative:

> *"What can I do with 2 acres – 2,000 or 3,000 meters of land? You can't do anything. In addition, we need water and effective water projects".*

The small scale of cultivation, and in the absence of effective farmers' cooperatives, makes it challenging to invest in modern technologies and development of modern agricultural techniques. The preservation of this scale of cultivation seems to be by design, as the farmers' union representative insisted that tobacco farming should remain in the hand of small farmers and not large property owners, hinting that the restriction of cultivation licenses to small farmers is a policy choice. It was noted that in several interviewees with various stakeholders, most notably the tobacco farmers union and Regie representatives, there is no interest in developing the tobacco farming sector and improving its efficiency.

Tobacco cultivation in Lebanon, relies mostly on family labor. In recent years, with the influx of Syrian refugee settling in mostly rural agricultural areas, and with the expansion of licensing to the Bekaa region, there was an increasing trend of hiring paid labor to do the work; however, this remains a limited practice due to the low returns:

> *"Tobacco agriculture needs many family members to work in it, for it to be profitable. Because if you must get workers, it won't be. In Beqaa, they don't work. They hire Syrian workers." (Regie representative).*

License renting is another common practice that was mentioned by several interviewees. With many license holders no longer practicing tobacco cultivation, they offer their licenses for a fee to others who are interested in producing larger quantities. Farmers collecting several licenses are then able to increase the quantities produced, exploiting economies of scale to make their production more efficient.

## Big tobacco companies

It should be noted that it was challenging to interview any representatives from the big tobacco companies. Despite the team's best effort to contact them, there was no publicly available contact information for any of the companies. After a significant search effort, two contacts for two separate companies were managed to be found. When contacted, they both declined to meet. Remaining out of the spotlight and evading public interaction seems to be the main approach adopted by these companies as noted below.

There is a long-standing collaboration between the Regie and prominent international tobacco companies, including Japan Tobacco International (JTI), Philip Morris, Imperial Brands, and British American Tobacco (BAT). This collaboration is multifaceted. In its capacity as the exclusive agency for tobacco distribution in Lebanon, RLTT purchases the finished tobacco products of the various brands sold by these companies, On the other hand, the Regie sells excess raw tobacco collected from farmers; RLTT purchases tobacco leaves from local farmers and, in return, sells these leaves to the international industry. This reciprocal agreement, as described by an interviewed researcher, establishes a mutual exchange.

> *"I don't buy international brands unless you buy our local leaves" (Researcher 1).*

The exclusive agency status of RLTT means that all companies trading in tobacco in the Lebanese market must go through RLTT. This long-standing relationship, dating back to 1935, is characterized by trust, with multinational companies sending products to RLTT, confident that they will be paid for their goods upon sale. This trust relationship is highlighted as a point of pride for RLTT, emphasizing the enduring and reliable nature of their partnerships:

> *"We are proud of something that is not mentioned in books: from 1935 until today, these multinational companies send us products trustfully. When we sell these products, we pay their price. You see the trust between us here" (Regie Representative).*

More recently, this cooperation has expanded to the production of international brands locally. The Regie has expanded its operations by manufacturing international brands since 2016, as indicated by the Regie representative in the interview. These agreements have led to the production of some of the world's leading tobacco brands at the Regie factory, which has expanded its operations to encompass almost 15 production lines. Cedars, the Regie signature brand, however, maintains its significant market share:

> *"Cedars constitute around 52 to 56% of the market share, but again this is also part of the economic crisis because many of the consumers shifted towards Cedars" (Researcher 1).*

Despite the evident impact of these partnerships, the nature of the relationship between the Regie and these tobacco giants remains ambiguous. The lack of clarity underscores the challenges in understanding the dynamics between the state-owned Regie and the influential international tobacco companies operating within Lebanon. Moreover, the reluctance of big tobacco companies to participate in interviews for our research study further complicates efforts to unveil the intricacies of their involvement, highlighting a persistent barrier to transparency and accountability within the Lebanese tobacco industry.

As shown in Fig 4, as of 2022, the Regie Libanaise de Tabacs et Tombacs (Regie) held a market share of more than 55% in the tobacco industry, up from 45% in 2017. The leading transnational tobacco company (TTC) in Lebanon was Philip Morris International (PMI), with a market share of nearly 15%. Following PMI were Japan Tobacco International (JTI) with over 11%, Imperial Brands with almost 10%, and British American Tobacco (BAT) with 6.5%.

## Distribution and wholesale

In Lebanon, tobacco distribution and wholesale operate within a structured framework overseen by the Regie and are subject to specific regulations. As explained by the Regie and by government representatives, the distribution network involves a hierarchy with heads of sales for each region, particularly in Beirut and its suburbs, where around 20 to 25 heads of sales manage the distribution process. A total of 668 retail and wholesale groups are authorized to distribute tobacco products at both retail and wholesale levels throughout Lebanon. Exclusive licenses to sell tobacco products are granted for a fee to these groups, and consequently, the distribution process primarily involves direct delivery of tobacco to sellers through Regie's stores, with retail sellers playing a central role [38]. Importantly, as highlighted by a governmental representative, private companies are restricted from directly importing tobacco and tombac into Lebanon without RLTT's approval. To initiate such imports, companies must submit a request to RLTT, specifying the brand, manufacturing details, and a financial study outlining costs and potential market prices.

The financial crisis and the associated chaos in the exchange rate in the past four years contributed to significant gains for tobacco wholesalers and retailers. RLTT adopted a multi-part pricing scheme based on multiple exchange rates that evolved over the duration of the crisis.

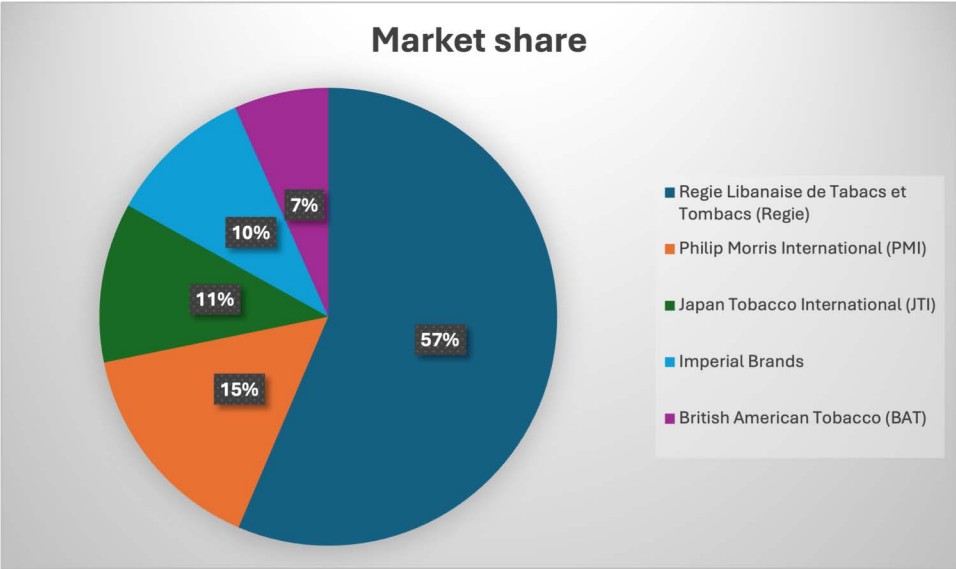

**Fig 4. Market share and leading brands (from Lebanon country profile, 2023 [ 30] based on Euromonitor International, company shares 2017—2022, published May 2023).**

Several researchers highlighted that, when considering the payment structure, wholesalers end up paying nearly 50% of the actual price if it was based on the real market exchange rate. This constituted another form of subsidization specific to the tobacco, while other essential goods and services in Lebanon do not benefit from similar subsidies.

*"If you do the math, you end up between the black-market exchange rate and the formula of RLTT, meaning that the wholesaler is buying a box of cigarettes at almost 50% of the actual price if it was being sold at the black-market exchange rate which is kind of, for me, is kind of a subsidization, because for food, medication and everything you don't have this kind of subsidy taking place anymore" (Researcher 1).*

## Public oversight and public policy

Lebanon, as a signatory to the WHO Framework Convention on Tobacco Control (FCTC), has been actively engaged in international efforts to curtail tobacco use globally. Notably, significant strides were made in tobacco control with the passage of Law No. 174 in August 2011, marking Lebanon's first tobacco control legislation [39]. It introduced comprehensive measures, including the prohibition of smoking in all indoor public spaces, a ban on tobacco advertising, promotion, and sponsorship, and the introduction of larger text warnings on tobacco products, with the potential for graphic health warnings in the future. It must be noted that in the past few years, lobbying by the tobacco industry has hindered the implementation of graphic health warnings. Additionally, Law 394 addressed warnings about the hazards of smoking. One of the interviewees from civil society, who was involved in the effort to introduce this law, noted that more than a decade after its passage, there has been a noticeable regression in its implementation, particularly regarding the enforcement of smoking bans in closed public spaces. Nonetheless, the interviewee noted that, despite this setback, there are some lasting effects of Law No. 174, most notably the ban on advertisements.

Law No. 174 was the result of a massive effort by civil society, academics, public health professionals and other stakeholders. The discussion with several interviewees, including a

parliamentarian, the regie representative and several participants in the advancement of this law provided important insight on the discussions with political actors and policymakers at that time. These insights should be informative for future strategies for policy making related to tobacco. There are four notable remarks: (1) Most policy makers, in both the executive and legislative branches, are open to discussions about tobacco control, they are aware of the public health concerns given the alarming trends in prevalence and significant youth consumption. (2) There is limited opposition from both Regie and stakeholders in private sector (including big tobacco companies, wholesalers, and the tourism industry) as long as the policy is focused on regulating access. (3) There is reluctance among policy makers, and opposition by Regie on raising taxes on tobacco products. (4) There is no attention from policy makers to the supply side of the tobacco industry or the need to reform the current structure. In short, the public policy approach concerning tobacco is to address the high prevalence by attempting to alter the demand of tobacco via regulations and campaigns aimed at consumer preferences.

Tobacco taxes in Lebanon have been persistently low. Moreover, the currency devaluation and the multiplicity of exchange rates have led to a decrease in the real value of the tax. Since the beginning of the crisis there has been reluctance to adjust the tobacco tax policy to reflect the new reality. According to the representative of the MOF, ongoing efforts have been dedicated to addressing tobacco-related issues. However, the multifaceted challenges within the country have hindered the prioritization of tobacco control. While certain accomplishments, such as the smoking ban, have been acknowledged, the implementation of these measures has fallen short. The representative highlighted the persistently weak enforcement of smoking-related laws including tobacco taxation policies:

> *"There has always been a weak implementation of the laws issued in relation to smoking. Today, taxes on tobacco are as follows: 2% excise tax on imported goods, 5% tax on tobacco products, 11% VAT and 108% domestic consumption. This rate is very low" (Representative of the MOF).*

Several explanations for the reluctance to increase taxes on tobacco were presented by the different interviewees. There are two views that summarize the approach by policy makers to taxation. The first was expressed by the Regie representative and other career administrators in the public sector. It highlights that the current approach leads to higher revenues to the treasury, and that higher taxes will lead to an increase in illicit trade.

> *"Why have the constraints of rigid taxes? Let's say I have a pack of cigarettes. The market today can bear that a pack of Cedars be sold at 100,000 LBP. I ask the minister to set its price at 100,000 LBP. Everything I can earn goes to the state. This is called tax. If the market can bear a price of 120,000 LBP, then I set it at that. While if you impose a tax, as they did in 1999, smuggling will increase, and we will lose. If I notice an increase in smuggling, I lower my prices to maintain the income. I can be flexible when it comes to that, so why would I impose constraints on myself with taxes?" (Regie representative).*

The second prevalent view is that policy makers in the government and the parliament have a conservative approach to legislation and policy change and prefer to maintain the status quo especially when they perceive the system to be working well. This was summarized by the parliamentarian representative by the following statement:

> *"Parliamentary discussions and public policy are often not based on facts or a comprehensive view of policy. They are rather based on perceptions. It is rare that scientific evidence is presented, and numbers are often an approximation or a point of view" (Parliament member).*

One of the main reasons why tobacco taxation and other tobacco-related policies are difficult to implement is that the governmental oversight over the tobacco industry in Lebanon is limited. In Principle, the MOF exercises control and oversight over RLTT, with a Government Commissioner delegated by RLTT who is supposed to monitor its works and financial affairs. However, this oversight is notably absent in practice. The MOF has limited interaction with RLTT, as the latter maintains direct connections with the minister, further complicating the regulatory landscape. In addition, the Ministry of Economy and Trade (MOET) has a role in regulating tobacco markets in Lebanon. This includes monitoring the market prices and ensuring that selling prices comply with the list issued by RLTT or the MoF in general, and monitoring the implementation of the law that bans smoking in public places, especially in restaurants, etc. This falls within the duties of the judicial police referring to the Consumer Protection Directorate in the MoET, which oversees ensuring the proper implementation of the law. The Ministry of Economy and Trade (MOET) plays a crucial role in fostering consumer awareness, aligning with the responsibilities delegated to the MOET, specifically under the purview of the Consumer Protection Directorate (Governmental representative from the MOET).

Despite the efforts of various entities, including the MOF, to implement public policies aimed at reducing tobacco consumption, the significant influence of the Regie on government decisions as well as its strong alliances with legislators often hinders their effectiveness (Alaouie et al. 2022). This is exemplified by the 2017 attempt to impose a 250 LBP (equivalent to 0.17 USD at the time) increase in cigarette prices as a domestic consumption fee. A representative from the MOF emphasized that while this increase was intended to generate revenue for the public treasury, it was ultimately used as profit by the RLTT, highlighting the extent of their control over government decision-making. This incident illustrates the ongoing conflict between balancing public health and public finance priorities with the financial interests of the RLTT.

## Permeability of Lebanon-Syria borders

Illicit trafficking of goods between Lebanon and Syria has existed since the establishment of the countries as independent nation states in the mid-1940s [40]. As reported by [21], the period of civil war witnessed a significant rise in tobacco smuggling, attributed to a local production decrease and inadequate state control over customs, leading to the influx of affordable contraband into the markets. Transnational Tobacco Companies (TTCs) took advantage of the chaos of the civil war to expand their market presence and supply tobacco to Lebanon or, in turn, facilitate distribution to neighboring countries like Syria, Jordan, and Turkey [21].

The direction of smuggling has often fluctuated based on comparative political and economic stability in Lebanon and Syria. Alaouie et al. (2022) [20] note the use of Lebanon as a gateway by TTCs for both legal and illegal trade of tobacco products in nearby nations. In earlier years, tobacco smuggling was more common from Syria to Lebanon due to difference in exchange rate [40,41]. There is a common agreement among most of the interviewees that in recent years, due to the war in Syria, the direction of smuggling inverted, with large quantities of tobacco products being smuggled from Lebanon to Syria:

> "In 2011, our trade margin reached its peak when the revolution in Syria started. Due to the war in Syria, the Regie in Syria stopped and the smuggling from here to Syria started" (Regie representative).

Smuggling continues between both countries due to the porous borders and lack of monitoring. However, currently, the smuggling trend has reversed, with tobacco moving from Lebanon to Syria:

> *"Because the product (Cedars) in Lebanon is very affordable, it's being "illicited" to the other countries including Syria, mainly Syria" (Researcher 1).*

> *"Just as there was Sayrafa, the exchange rate for dollars platform, there was one for RLTT, at half of the real value of the dollar. If the price of a box of Cedars is sold at 300 dollars, its actual price is 150 dollars. What happened? It started to get smuggled to Syria because it was cheap. Our products were smuggled, yet our profits increased instead of decreasing" (Regie representative).*

Tobacco smuggling from Lebanon to other countries, particularly Syria, presents a complex interplay of economic factors and regulatory challenges. According to the 2023 Global Organized Crime Index for Lebanon, as reported by the Global Initiative Against Transnational Organized Crime, the illicit tobacco trade makes up approximately one-fourth of the entire tobacco market. Lebanon, known for its affordable tobacco products, becomes a focal point for illicit trade, with the smuggling route traversing through Turkey and Syria. Interestingly, it is believed by many of the stakeholders and researchers that were interviewed in this study that the Regie is well aware of the smuggling activities occurring beyond the country's borders. Contrary to expectations, the RLTT does not actively intervene or halt these operations. According to an interviewed researcher, the RLTT's decision to allow smuggling is strategic, as it serves as a significant source of revenue, which, in turn, contributes to sustaining their production activities. According to the information available on their official website, the Regie claims to be actively engaged in anti-smuggling operations. This involves market monitoring, combating the smuggling of tobacco and tombac throughout Lebanon, conducting raids on factories producing counterfeit tobacco products, and inspecting stores selling forged or smuggled items [42]. RLTT also oversees the free zone at Beirut port, ensuring the secure unloading and transportation of tobacco fabricated in the free zone to destinations like the airport or other ports [42].

It has been expected for the issue of tobacco taxation to add another layer to the situation's complexity. Researchers highlighted that, historically, when advocates pushed for increased tobacco taxes to curb youth consumption and overall tobacco use, the RLTT, MOF and customs authorities argued against it. Their primary concern was that higher taxes would lead to an escalation in smuggling activities, particularly from Syria. This has been the main argument for many tobacco industries and tax opponents in other countries. While it is commonly acknowledged that the potential for tobacco smuggling can constrain the implementation of higher tobacco tax rates, entities opposing increases in tobacco taxes often tend to exaggerate both the extent and the risks associated with smuggling [16]. Estimates indicate that the increase in smuggling due to higher taxation is marginal and often overstated.

## Discussion and policy recommendations

The findings presented in the previous section provide a general overview of the tobacco industry in Lebanon. While a first read of the evidence portrays a complex picture of many stakeholders with intricate relationships that are in many ways asymmetric, a broader and more focused consideration of the context shows a more trivial political economy story centralized around monopolization and conflicting interests.

The following discussion elaborates on three major themes that were identified as key features of the tobacco industry in Lebanon and provides policy recommendations to

address these issues. The recommendations made below are based on the assumed aim of economic welfare improvement by attaining public policy goals in both public health and public finance.

## Central role of the Regie as a state-owned monopoly

The central role of the Regie as a state-owned monopoly in the tobacco industry leads to conflict with potential key public policy objectives. This is evident when it comes to the tension between maximizing Regie's revenues from tobacco sales in the local market and implementing an optimal tax policy. What constitutes an optimal tobacco tax policy objective could be subject to further discussion. Nonetheless, it is evident from discussions that revenues to the treasury and reduction of health costs associated with tobacco consumption should be key determinants of the policy ([43,44]).

The view that the Regie's distribution of dividends to the Ministry of Finance resolves this conflict is not accurate. As discussed above, the Regie allocates a significant amount of revenue at the discretion of its independent management, with limited intervention from any public body, including the Ministry of Finance. This discretionary spending is not necessarily coordinated with public objectives set by the Ministry of Finance or the government more broadly. A second source of this contradiction stems from the Regie de facto operating as a supplier of tobacco (mostly cigarettes) to a larger regional market, not just the Lebanese market. This is particularly evident given indications of significant smuggling activity toward Syria and the Regie's increased cigarette production in recent years, which may suggest a response to this market demand. Licensing deals agreed upon with the big tobacco companies to produce international brands in Lebanon, seems to be partially motivated by this unofficial market expansion. The low tobacco tax rate policy currently adopted in Lebanon increases the competitiveness of Regie-produced cigarettes in regional markets [45].

In addressing the challenges within the tobacco industry, several key policy recommendations emerge. Firstly, restructuring the industry is paramount, necessitating the liberalization of the sector and the dissolution of the Regie monopoly and monopsony through horizontal and vertical integration. Secondly, implementing an excise-specific tax on domestically manufactured tobacco goods is crucial, with the proceeds directed entirely to the government's treasury [46]. Lastly, optimizing ad valorem and other sales taxes on locally traded tobacco items in alignment with market dynamics is essential to maximize revenue generation for the treasury [47]. These recommendations collectively aim to foster a more competitive and revenue-efficient tobacco market while ensuring effective government oversight and fiscal gains.

## There is no clear national strategy on tobacco that translates into actionable public policy

Just as with taxation and tobacco price setting, the central role of the Regie as a state-owned monopoly hindered the different public policy actors from playing a role in regulating the tobacco industry. The only active policy on tobacco spurs public health concerns. The central feature of this policy focuses on incentivizing consumption habits through behavioral means. Discussions with representatives from various public entities (other than the Ministry of Finance) demonstrated a lack of interest in intervention in any policy related to regulating the tobacco industry, recognizing it as Regie's territory. This passive approach towards a major industry in the country is demonstrative of a limited understanding of the regulatory role of governmental bodies in defining, promoting, and devising strategies for social and economic progress.

The current circumstances in the country with the ongoing crisis that started in October 2019 poses significant challenges to policy making, but it also presents a significant opportunity for extensive structural changes that would have been otherwise deemed excessive in non-crisis times. Recent significant policy changes, such as the introduction of a national antitrust and competition law (number 281 of 2022), is one example of the changing approach to public policy but is also an attestation to a changing mentality among the general population on social issues.

To address these issues, several key policy recommendations are proposed. Firstly, it is suggested to expand the existing antitrust law to encompass the tobacco industry, with specific amendments aimed at ensuring fair competition once the industry undergoes liberalization. Additionally, legislative measures should be enacted to allow for earmarked taxes and fees on tobacco products ([48,49]).

Two significant fees are identified as potentially beneficial in the current context: a national ad valorem tax designed to fund national healthcare providers such as the Ministry of Public Health and the "Maternity and Sickness Fund" of the National Social Security, among others; and a local fee on tobacco sales within municipal jurisdictions, intended to finance initiatives aimed at mitigating the local environmental impact of smoking. These policy recommendations aim to address both competition concerns within the industry and the broader societal costs associated with tobacco consumption.

## Tobacco farmers' interest and economic survival is a key component of any reform of the tobacco industry

The evidence demonstrates that tobacco cultivation in Lebanon is inefficient. The prospect that this practice will become economically sustainable in the future is doubtful, an opinion shared by most interviewed experts, including the tobacco farmers union member. Therefore, from an economic efficiency perspective, there is no rationale for the tobacco cultivation subsidy program. Nonetheless, the subsidy and the community development programs funded by the Regie in tobacco producing districts constitute a substantive transfer with many beneficiaries. As noted in the discussion on tobacco farming, tobacco cultivation is a family practice relying on the participation of most family members. These families are often low-income families, for whom tobacco cultivation is a part-time, seasonal, occupation that brings in some needed additional income and cash flow. Moreover, the two governorates with the bulk of tobacco cultivation, Akkar and Nabatieh, have the highest rates of poverty in the country [50]. Any changes to the current tobacco industry structure could have a significant negative impact on these farmers, making such proposals politically undesirable. As such, any reform should take into consideration a form of compensation for the current tobacco farming license holders. It should be noted that several of the challenges facing tobacco farmers are common to the general agricultural sector in Lebanon. Encouraging alternative crops is a policy often discussed and promoted by experts and policy makers, however, several recent experiences have seen limited success.

Given the potential impacts of subsidy removal and the restructuring of Lebanon's tobacco industry on farmers, it is imperative to safeguard the interests of this vulnerable group through targeted policy interventions. Key recommendations include firstly, abolishing the current cultivation licensing framework and offering financial compensation to existing license holders. This compensation could take the form of either a direct cash buyback of licenses or ownership shares in the restructured tobacco industry, ensuring a fair transition. Secondly, empowering the Ministry of Agriculture to devise a program that supports farmers in transitioning away from tobacco cultivation towards crops that contribute positively to Lebanon's food security landscape. This initiative would not only benefit farmers but

also bolster the nation's agricultural diversity. Lastly, formulating a comprehensive national agricultural policy that prioritizes the development of agroindustry as a viable and lucrative alternative for farmers, thereby fostering economic sustainability and resilience within the agricultural sector. These recommendations collectively aim to mitigate the adverse effects of industry restructuring on farmers while promoting long-term agricultural viability and food security in Lebanon.

## Conclusion

Examining Lebanon as a case study is not only pertinent but also imperative for understanding and addressing complex issues within the global tobacco industry. This study addresses a significant gap in the literature by exploring the political economy of the tobacco supply chain in Lebanon—a topic that has been underexplored despite its critical importance for effective tobacco control strategies, particularly taxation policies. The research question at the core of this paper examines how the existing market structure influences the power dynamics and interactions among stakeholders within the Lebanese tobacco industry. Through an exploratory qualitative approach, we mapped the tobacco supply chain, capturing the complex relationships among key players, including the state-owned tobacco monopoly, farmers, private companies, and policymakers.

While Lebanon's high prevalence of smoking sets the stage for investigation, the significance of this research extends far beyond statistical figures. The country's operation of a state-owned tobacco monopoly, the Regie Libanaise de Tabacs et Tombacs, presents a unique opportunity to delve into the intricate dynamics of monopolistic practices within the tobacco sector. This understanding is crucial, especially considering the Monopoly-Oriented Endgame Models (MOEM) that propose strategies for drastically reducing smoking prevalence, including shifting the supply of tobacco to some form of monopoly. This exploration not only enriches our understanding of Lebanon's tobacco industry but also offers insights into potential strategies for curbing smoking rates that could be applicable in diverse global contexts.

Moreover, the examination of Lebanon's tobacco supply chain provides a comprehensive view of production practices, distribution networks, marketing strategies, and the substantial influence wielded by multinational tobacco companies on consumption patterns. Unraveling these aspects is crucial for formulating effective tobacco control policies, particularly considering the industry's adeptness at influencing policy-making bodies against implementing stringent measures. By delving deep into the operations of the Regie and mapping the interactions among stakeholders, this research contributes valuable insights into the complexities of the tobacco industry's political economy.

Our findings reveal two critical issues: first, the misalignment between the profit-driven objectives of the monopoly and the multifaceted goals of public policymakers, which include improving public health, environmental outcomes, and maximizing public revenue; and second, the co-dependency between the monopoly and subsistence farming, where the monopoly gains social legitimacy by subsidizing and sustaining inefficient farming practices.

Theoretically, this study contributes to the broader understanding of how market structures in politically sensitive industries like tobacco can shape stakeholder power dynamics, which in turn affects policy outcomes. Practically, the research offers valuable insights for policymakers in Lebanon and similar contexts, providing a blueprint for designing and implementing more effective tobacco control measures. By highlighting the intricate dynamics within the tobacco supply chain, this study lays the groundwork for future policy discussions and reforms aimed at reducing tobacco consumption and its associated harms.

Beyond Lebanon's borders, this study's findings and policy recommendations hold relevance for similar settings globally, especially within the Eastern Mediterranean region, where WHO projections indicate a worrying rise in smoking prevalence [13]. The innovative use of an exploratory qualitative approach not only fills a significant gap in the literature regarding the political economy of tobacco but also sheds light on crucial issues such as monopolization, the absence of a national tobacco strategy, and inefficient cultivation practices. These insights are not only actionable for policymakers in Lebanon but also serve as a blueprint for crafting effective tobacco control policies in other countries facing similar challenges. Thus, this research serves as basis for guiding efforts to combat smoking and its associated socioeconomic impacts on a broader scale.

The study's strengths lie in its innovative use of an exploratory qualitative approach to comprehensively examine the tobacco supply chain in Lebanon, filling a significant gap in the literature regarding the political economy of tobacco. By collecting innovative qualitative data, the study provides valuable insights into complex stakeholder interactions and underlying interests. The relevance of Lebanon as a case study, given its high smoking prevalence, state-owned tobacco monopoly (the Regie), and ongoing financial crisis, adds weight to the findings and allows for potential generalization to other contexts. The study's policy recommendations, addressing issues such as monopolization, lack of a national tobacco strategy, and inefficient cultivation practices, offer actionable insights for policymakers.

## Limitations of the study

The study has several limitations, including potential challenges in generalizing its findings to countries with significantly different supply chain structures in the tobacco industry. Additionally, relying solely on qualitative data may limit the depth of quantitative analysis needed to fully assess the economic impacts of proposed policy changes. Another limitation is the lack of participation from major tobacco firms in the study. On one hand, this is a limitation in terms of the data, as it leaves one side of the picture unexplored. On the other hand, given that this study focuses on the political economy of tobacco, the refusal of big tobacco firms to participate in the interviews may suggest a reluctance to be transparent, possibly indicating an interest in withholding information about the extent of their benefits from the current status quo.

There is also a possibility of bias in the data collected, given the strong political ties of the tobacco industry in Lebanon which may have influenced the responses of certain stakeholders, potentially introducing bias. Moreover, the feasibility and practicality of implementing the suggested policy interventions, especially within Lebanon's complex political and economic landscape, may pose significant hurdles that need careful consideration.

Future research should triangulate qualitative data with updated quantitative data to assess the impact of policy changes. This includes updated estimates of demand elasticities, tax simulation exercises, impact on revenues to the public treasury, distributional impact of the tax, and impact on prevalence [51].

## Supporting information

**S1 File. Semi-structured interview guide (Key informants).**
(DOCX)

## Author contributions

**Conceptualization:** Ali Chalak, Ali Abboud, Joanne Haddad, Mariam Radwan.

**Data curation:** Ali Chalak, Ali Abboud, Mariam Radwan.

**Formal analysis:** Ali Chalak, Ali Abboud, Joanne Haddad, Mariam Radwan.

**Funding acquisition:** Ali Chalak, Ali Abboud, Mariam Radwan.

**Investigation:** Ali Chalak, Ali Abboud, Mariam Radwan.

**Methodology:** Ali Chalak, Ali Abboud, Joanne Haddad, Mariam Radwan.

**Project administration:** Ali Chalak, Ali Abboud, Mariam Radwan.

**Resources:** Ali Chalak, Ali Abboud, Mariam Radwan.

**Software:** Ali Chalak, Ali Abboud, Mariam Radwan.

**Supervision:** Ali Chalak, Ali Abboud, Mariam Radwan.

**Validation:** Ali Chalak, Ali Abboud, Mariam Radwan.

**Visualization:** Ali Chalak, Ali Abboud, Joanne Haddad, Mariam Radwan.

**Writing – original draft:** Ali Chalak, Ali Abboud, Joanne Haddad, Mariam Radwan.

**Writing – review & editing:** Ali Chalak, Ali Abboud, Joanne Haddad, Mariam Radwan.

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
