## [Decision Letter · Decision Letter 0]

9 Jul 2024

PONE-D-24-14365A Political Economy of the Tobacco Supply Chain in an Eastern Mediterranean Country: The Case of Lebanon

PLOS ONE

Dear Dr. Chalak,

Thank you for submitting your manuscript to PLOS ONE. After careful consideration, we feel that it has merit but does not fully meet PLOS ONE’s publication criteria as it currently stands. Therefore, we invite you to submit a revised version of the manuscript that addresses the points raised during the review process.

We look forward to receiving your revised manuscript.

Kind regards,

Muhammad Hashim, PhD

Academic Editor

PLOS ONE

 [The project is funded by the University of Illinois at Chicago’s (UIC) Institute for Health Research and Policy to conduct economic research on tobacco in Lebanon. UIC is a partner of the Bloomberg Initiative to Reduce Tobacco Use. The views expressed in this paper cannot be attributed to, nor do they represent, the views of the American University Beirut, UIC, the Institute for Health Research and Policy, or Bloomberg Philanthropies.].  

*Comments from the Journal: * One or more of the reviewers has recommended that you cite specific previously published works. Members of the editorial team have determined that the works referenced are not directly related to the submitted manuscript. As such, please note that it is not necessary or expected to cite the works requested by the reviewer.

Reviewers' comments:

Reviewer's Responses to Questions

**Comments to the Author**

1. Is the manuscript technically sound, and do the data support the conclusions?

Reviewer #1: Partly

Reviewer #2: Partly

2. Has the statistical analysis been performed appropriately and rigorously?

Reviewer #1: N/A

Reviewer #2: Yes

3. Have the authors made all data underlying the findings in their manuscript fully available?

Reviewer #1: Yes

Reviewer #2: No

4. Is the manuscript presented in an intelligible fashion and written in standard English?

Reviewer #1: Yes

Reviewer #2: No

5. Review Comments to the Author

Reviewer #1: The idea of the paper is interesting. There are concerns as follows:

The rationale behind the chosen methodology with support from the academic literature.

The literature review section needs further attention. Papers worked on policy suggestions would be interesting to review.

For example:

Green subsidy modes and pricing strategy in a capital-constrained supply chain. Transportation Research Part E: Logistics and Transportation Review, 136, p.101885.

Alternative governmental carbon policies on populations of green and non-green supply chains in a competitive market. Environment, Development and Sustainability, 25(5), pp.4139-4172.

Joint impact of CSR policy and market structure on environmental sustainability in supply chains. Computers & Industrial Engineering, 185, p.109654.

This will, in turn, link the work to existing quantitative works in the literature that work on optimal policies. Such that will strengthen future work directions.

The conclusion section will need further revision accordingly.

Reviewer #2: Reviewer comments

Introduction

1. The introduction needs to be improved. Please, use more recent literature including those in 2024.

2. The problem or gap is not well argued out. You must properly pitch your study in extant literature.

3. Besides, it is not enough to say “…far less attention has been dedicated to examining the political economy of tobacco supply chains and associated, tobacco industry practices, as important as such an understanding is to the design and implementation of effective taxation policies.” And so what? What is the effect of the above? You must make a good case for your study.

4. Why have “background?” I suggest you find a way to clearly merge the background with the introduction.

5. Please condense your objective into a research question.

6. You must also clearly state the contributions of your study….unlike previous studies.

7. Don’t make the introduction too long…it makes the reading boring.

Methodology

1. Why do you have “methodology and sample description” ? isn’t sample description part of the methodology? Refer to the structure of similar papers in this journal and properly label your captions under the methodology.

2. You must justify the choice of each technique. For example, why you used qualitative or purposive sampling?

3. How was the analysis done? The present status is too shallow. Did you use software or manual? And how the themes discovered? Did you do immersion? Coding? Refer to Kwarteng et al., (2022) for more insight into thematic analysis. This part of the work is weak. This important to guarantee transparency and trust in your results.

4. Also refer to Kanda et al. (2021) and Kwarteng et al., (2022) to see how they presented their results. Use insights from these papers to better your presentation of results.

5. Some aspects of the discussions were done from the perspective of extant literature (What we already know) that is good. This must be replicated through the discussions (where applicable).

6. I discovered that there was no theoretical background in the study. Neither were the discuses done in line with any specific theory. Anyway, is the study an inductive or deductive one? Don’t you think it would be nice to mention how your study supports or disprove some theories. OR if you are doing an inductive research, we must state in your contribution how you are contributing to the development of theory. This is seriously lacking in your study.

Conclusion

1. The conclusion must be well written. It must capture the gap, research question, methods and summary of the findings

2. Also provide the theoretical and practical contribution of study

3. What are the limitation and directions for future studies?

4. Each of the above (2 and 3) should be subtopics under the conclusion

5. The language must be improved. Get a language expert to check the grammar

6. Ensure that all citations are properly referenced.

z

6. PLOS authors have the option to publish the peer review history of their article (what does this mean? ). If published, this will include your full peer review and any attached files.

Reviewer #1: No

Reviewer #2: No

---

## [Author Response · Author response to Decision Letter 1]

23 Aug 2024

Responses to Reviewers’ Comments for Manuscript PONE-D-24-14365

A Political Economy of the Tobacco Supply Chain in an Eastern Mediterranean Country: The Case of Lebanon

By Ali Abboud, Ali Chalak, Joanne Haddad, Mariam Radwan

Dear Professor Muhammad Hashim,

We are very grateful to you as editor and to the anonymous reviewers for their comments, which were thoughtful, fair, and very helpful. We believe we have put together a fully responsive revision. In general, we agreed with the vast majority of comments and amended the paper in the directions suggested. We hope you will agree that the substance and packaging of the analysis has been considerably improved by the process of review and revision.

Throughout, we use boxes to identify the comments from the reviewer and plain text

capture our responses. Except where made explicit, page, table and figure numbers refer

to those in the new version.

We start by providing a summary of the major changes, followed by a point-by-point

response to the issues and suggestions raised by the referees.

Sincerely,

Ali Abboud, Ali Chalak, Joanne Haddad, Mariam Radwan

Major Changes

1. Justifying Methodological Choices

To address the recurrent comment from both anonymous reviewers regarding the rationale behind our chosen methodology and the justification of each technique, we have made significant revisions to the manuscript. Specifically, we have edited the methodology section to thoroughly justify our methodological choices and explain the rationale behind each technique used.

Summary of Changes:

1. We have provided a detailed explanation for using qualitative methods, supported by relevant academic literature and tobacco-related studies.

2. We have included a rationale for employing different techniques such as purposive sampling, induct

3. ive analysis approach, and thematic analysis, referencing scholarly sources to strengthen our justification.

4. We have referred to the suggested qualitative papers and have provided more in-depth description of the analysis process. Each step of the analysis is now mentioned in the methodology section to ensure more rigor and transparency of our results.

A detailed overview of these changes is provided in our response to the reviewers.

2. Clarifying the Research Gap, Objectives, and Contributions

To address one major comment raised by anonymous reviewer #2 regarding the need to better justify the study's relevance, clearly define the research question, and distinctly state the contributions of the study in comparison to previous research, we have made significant changes to the manuscript to address these concerns.

Specifically, we have:

1. We provided a comprehensive justification for the study's relevance, supported by a thorough review of the existing literature. In our discussion, we highlight how this study complements and adds to the existing body of work by addressing two significant issues in the Lebanese context: the existence of a profit-driven monopoly that conflicts with public policymakers' broader goals, including public revenue, health, and environmental improvements, and the inefficient subsidizing of subsistence farming, which enables the monopoly to gain legitimacy as a social provider. These dynamics underscore the complexities of Lebanon's tobacco supply chain and their importance for informed policymaking.

2. We Clearly defined the research question, ensuring it is concise and directly addresses the identified gap.

3. We articulated the unique contributions of the study, highlighting how it differs from and builds upon previous research as mentioned in point 1, as well as in our discussion of the importance of the qualitative approach adopted which offers a unique perspective that was not inspected in earlier studies on the tobacco supply chain in Lebanon. The qualitative approach is crucial as it provides an in-depth exploration of power dynamics among stakeholders in Lebanon's tobacco supply chain, which previous studies have overlooked. Additionally, it compensates for the lack of reliable quantitative data, offering valuable insights that are essential for understanding similar complex political economy relationships in other contexts.

A detailed overview of these changes is provided in our response to reviewer #2.

Authors’ Response to Reviewer 1

Response: We thank the reviewer for their positive positive feedback. To address this concern, we have undertaken significant revisions to the manuscript to better explain the rationale behind our chosen methodology. In particular, we have amended the introduction, methodology and conclusion sections.

We now write in p. 9 “According to Mathie and Camozzi (2005), qualitative research is especially useful for “politically or socially sensitive topics”, such as the dynamics of tobacco production and products smuggling. Furthermore, qualitative methods have been widely used in tobacco-related research. Newly published research delves into the governmental strategies and perspectives on tobacco control and its alternatives in Malawi. It involved semi-structured interviews with stakeholders working in the tobacco sector (Lencucha et al., 2020). Moreover, a recent study aimed to explore challenges facing tobacco control policies in Indonesia by conducting interviews with national tobacco control experts including academics, community organizations, and government officials (Astuti, Assunta, & Freeman, 2020). Another qualitative descriptive study in Australia focused on identifying key challenges hindering the progress in tobacco control by interviewing 31 individuals from various sectors involved in tobacco control. Similarly, a study aiming to understand the dynamics of the waterpipe industry also utilized semi-structured interviews with representatives from various waterpipe companies (Jongenelis, 2022). In many studies, the use of qualitative methodology has been considered a strength given the richness of data that it provides regarding the tobacco industry especially when official documents that could shed light on the market dynamics are lacking”.

Going beyond support from the academic literature, we have also discussed the importance of the qualitative approach adopted which offers a unique perspective that was not inspected in earlier studies on the tobacco supply chain in Lebanon. The qualitative approach is crucial as it provides an in-depth exploration of power dynamics among stakeholders in Lebanon's tobacco supply chain, which previous studies have overlooked. Additionally, it compensates for the lack of reliable quantitative data, offering valuable insights that are essential for understanding similar complex political economy relationships in other contexts.

We now write in p.5 “The adoption of a qualitative approach offers a unique perspective that was not inspected in earlier studies on the tobacco supply chain in Lebanon. First, it allows a detailed exploration of relationships of interests and power between different stakeholders (The monopole, farmers, private companies, regulators, and legislators). Second, given the limited availability and reliability of quantitative data on tobacco in Lebanon, and the deliberate ambiguity in public reports, the collected qualitative data offers crucial amount of information that would otherwise be unavailable. As these issues are not unique to the tobacco sector in Lebanon, the methodological approach adopted in this paper offers a blueprint to analysing political economy relationship in other context with similar structural complexities and data limitations.”

Response: We thank the reviewer for their suggestions regarding the inclusion of papers that focus on policy and would enhance the literature review section. While we highly appreciate the reviewer’s input, we have decided not to cite the specific works suggested, as we, along with the journal and editorial team, believe they are not directly related to our submitted manuscript. Instead, following the reviewer’s suggestion, we have cited recent policy-related papers that specifically address the most effective approaches for reducing tobacco smoking prevalence. In particular, we have added the following citations:

Burki S, et al. The Economics of Tobacco and Tobacco Taxation in Pakistan. Paris: International Union Against Tuberculosis and Lung Disease; 2013.

Foucade AL, et al. The potential for using alcohol and tobacco taxes to fund prevention and control of noncommunicable diseases in Caribbean Community countries. Revista panamericana de salud publica = Pan American journal of public health. 2018;42:e192-e. doi: 10.26633/RPSP.2018.192

Goodchild M, et al. Tobacco control and Healthy China 2030. Tobacco control. 2019;28(4):409-13. doi: 10.1136/tobaccocontrol-2018-054372

Levy DT, et al. Application of the Abridged SimSmoke model to four Eastern Mediterranean countries. Tobacco control. 2016;25(4):413-21. Epub 2015/06/16. doi: 10.1136/tobaccocontrol-2015-052334

Levy DT, et al. England SimSmoke: the impact of nicotine vaping on smoking prevalence and smoking-attributable deaths in England. Addiction (Abingdon, England). 2021;116(5):1196-211. Epub 20201008. doi: 10.1111/add.15269

Sánchez-Romero LM, et al. The Kentucky SimSmoke Tobacco Control Policy Model of Smokeless Tobacco and Cigarette Use. International journal of health policy and management. 2020. Epub 20201027. doi: 10.34172/ijhpm.2020.

Tesche J, et al. Measuring the effects of the new ECOWAS and WAEMU tobacco excise tax directives. Tobacco control. 2021;30(6):668-74. Epub 2020/09/30. doi: 10.1136/tobaccocontrol-2020-055843

Cho, A., Lim, C., Sun, T., Chan, G., & Gartner, C. (2024). The effect of tobacco tax increase on price‐minimizing tobacco purchasing behaviours: A systematic review and meta‐analysis. Addiction.

Additionally, and following reviewer # 2 suggestion, we have improved our introduction to cite more recent literature including that in 2024. In particular, we have added the following citations:

Soriano, J. B., Kendrick, P. J., Paulson, K. R., Gupta, V., Abrams, E. M., Adedoyin, R. A., ... & Moradi, M. (2020). Prevalence and attributable health burden of chronic respiratory diseases, 1990–2017: a systematic analysis for the Global Burden of Disease Study 2017. The Lancet Respiratory Medicine, 8(6), 585-596.

Perez-Warnisher, M. T., de Miguel, M. P. C., & Seijo, L. M. (2018). Tobacco use worldwide: legislative efforts to curb consumption. Annals of global health, 84(4), 571.

World Health Organization. (2024). Tobacco use declines despite tobacco industry efforts to jeopardize progress. Pan American Health Organization. Retrieved from https://www.paho.org/en/news/16-1-2024-tobacco-use-declines-despite-tobacco-industry-efforts-jeopardize-progress.

Research and Markets. (2024). Global tobacco market report by type, and region 2024-2032. Retrieved from https://www.researchandmarkets.com/reports/5993422/global-tobacco-market.

El-Awa F, Bettcher D, Al-Lawati JA, Alebshehy R, Gouda H, Fraser CP. The status of tobacco control in the Eastern Mediterranean Region: progress in the implementation of the MPOWER measures. East Mediterr Health J. 2020 Jan 30;26(1):102-109. doi: 10.26719/2020.26.1.102. PMID: 32043552.

Saad RK, Maiteh A, Nakkash R, Salloum RG, Chalak A, Abu-Rmeileh NME, Khader Y, Al Nsour M. Monitoring and Combating Waterpipe Tobacco Smoking Through Surveillance and Taxation. JMIR Public Health Surveill. 2023 Mar 23;9:e40177. doi: 10.2196/40177. PMID: 36951907; PMCID: PMC10132023.

Robert West, Tobacco control: present and future, British Medical Bulletin, Volume 77-78, Issue 1, 2006, Pages 123–136, https://doi.org/10.1093/bmb/ldl012

U.S. National Cancer Institute and World Health Organization. The Economics of Tobacco and Tobacco Control. National Cancer Institute Tobacco Control Monograph 21. NIH Publication No. 16-CA-8029A. Bethesda, MD: U.S. Department of Health and Human Services, National Institutes of Health, National Cancer Institute; and Geneva, CH: World Health Organization; 2016.

Chaloupka, F. J., Yurekli, A., & Fong, G. T. (2012). Tobacco taxes as a tobacco control strategy. Tobacco control, 21(2), 172-180.

Azagba, S., Ebling, T., & Korkmaz, A. (2024). Beyond the smoke: Historical analysis of the revenue implications of state cigarette tax policies, 1989 to 2019. International Journal of Drug Policy, 127, 104408.

Response: Following this round of revisions and considering the comments from the two anonymous reviewers, the conclusion of the manuscript has been substantially revised. We have addressed the feedback on the methodology, particularly by explaining the rationale behind it. Additionally, we have clarified the research question, objectives, and the gap our study addresses in the literature. We have also included the limitations and strengths of this study in the conclusion section, following the recommendation of Reviewer 2.

All modifications have been made in track changes mode.

Authors’ Response to Reviewer 2

Introduction

Response: We thank the reviewer for this suggestion. We have amended the introduction to incorporate more recent literature. Specifically, following reviewer 1's suggestion, we have cited recent policy-related papers that address the most effective approaches for reducing tobacco smoking prevalence. These new citations are listed in our response to reviewer 1's second comment and are not repeated here.

Following this suggestion, we have added the following citations:

- Soriano, J. B., Kendrick, P. J., Paulson, K. R., Gupta, V., Abrams, E. M., Adedoyin, R. A., ... & Moradi, M. (2020). Prevalence and attributable health burden of chronic respiratory diseases, 1990–2017: A systematic analysis for the Global Burden of Disease Study 2017. The Lancet Respiratory Medicine, 8(6), 585-596.

- Perez-Warnisher, M. T., de Miguel, M. P. C., & Seijo, L. M. (2018). Tobacco use worldwide: Legislative efforts to curb consumption. Annals of Global Health, 84(4), 571.

- World Health Organization. (2024). Tobacco use declines despite tobacco industry efforts to jeopardize progress. Pan American Health Organization. Retrieved from [https://www.paho.org/en/news/16-1-2024-tobacco-use-declines-despite-tobacco-industry-efforts-jeopardize-progress](https://www.paho.org/en/news/16-1-2024-tobacco-use-declines-despite-tobacco-industry-efforts-jeopardize-progress).

- Research and Markets. (2024). Global tobacco market report by type, and region 2024-2032. Retrieved from [https://www.researchandmarkets.com/reports/5993422/global-tobacco-market](https://www.researchandmarkets.com/reports/5993422/global-tobacco-market).

- El-Awa, F., Bettcher, D., Al-Lawati, J. A., Alebshehy, R., Gouda, H., & Fraser, C. P. (2020). The status of tobacco control in the Eastern Mediterranean Region: Progress in the implementation of the MPOWER measures. Eastern Mediterranean Health Journal, 26(1), 102-109. doi: 10.26719/2020.26.1.102. PMID: 32043552.

- Saad, R. K., Maiteh, A., Nakkash, R., Salloum, R. G., Chalak, A., Abu-Rmeileh, N. M. E., & Khader, Y. (2023). Monitoring and combating waterpipe tobacco smoking through surveillance and taxation. JMIR Public Health and Surveillance, 9, e40177. doi: 10.2196/40177. PMID: 36951907; PMCID: PMC10132023.

- West, R. (2023). Tobacco control: Present and future. British Medical Bulletin, 77(1), 123–136. doi: 10.1093/bmb/ldl012.

- U.S. National Cancer Institute and World Health Organization. (2016). The economics of tobacco and tobacco control. National Cancer Institute Tobacco Control Monograph 21. NIH Publication No. 16-CA-8029A. Bethesda, MD: U.S. Department of Health and Human Services, National Institutes of Health, National Cancer Institute; and Geneva, CH: World Health Organization.

- Chaloupka, F. J., Yurekli, A., & Fong, G. T. (2012). Tobacco taxes as a tobacco control strategy. Tobacco Control, 21(2), 172-180.

- Azagba, S., Ebling, T., & Korkmaz, A. (2024). Beyond the smoke: Historical analysis of the revenue implications of state cigarette tax policies, 1989 to 2019. International Journal of Drug Policy, 127, 104408.

---

## [Decision Letter · Decision Letter 1]

31 Jan 2025

PONE-D-24-14365R1

A Political Economy of the Tobacco Supply Chain in an Eastern Mediterranean Country: The Case of Lebanon

PLOS ONE

Dear Dr. Chalak,

Thank you for submitting your manuscript to PLOS ONE. After careful consideration, we feel that it has merit but does not fully meet PLOS ONE’s publication criteria as it currently stands. Therefore, we invite you to submit a revised version of the manuscript that addresses the points raised during the review process.

We look forward to receiving your revised manuscript.

Kind regards,

Fabrizio Ferretti, PhD

Academic Editor

PLOS ONE

Journal Requirements:

Additional Editor Comments:

some statements have been presented throughout the manuscript without the addition of references or primary literature to support them, and it is unclear whether the information was obtained through the interviews or from other sources:

1. "In practice, however, there is only minimal interaction and active oversight, which results in the Regie being an independently administered monopoly operating with very limited oversight by any elected or governmental body" : Reference not provided

2. "The Regie has the exclusive right to provide licenses to farmers and wholesalers, to purchase locally produced tobacco leaves from Lebanese farmers, to purchase and import raw tobacco material, to regulate and set tobacco products standards, as well as to manufacture and distribute tobacco products": Reference not provided

3 "According to a researcher involved in tobacco monopolies and taxation policies research, it is estimated that 5 to 6 billion dollars were transferred to the treasury between 1990 and 2020": Reference not provided. Alternatively, a clarification from the authors of whether this researcher was part of the interviews.

4. "The Regie runs a subsidy program for tobacco farmers. The subsidy comes in the form of a negotiated price that is 60 to 100 percent higher than the global market price for tobacco leaves. Between 1994 and 2021, the subsidy cost was 1.032 billion USD." : Reference not provided.

5. "The Regie has expanded its operations by manufacturing international brands since 2016." : Reference not provided.

6. "More than a decade after the passage of the law, there has been a noticeable regression in its implementation, especially when it comes to enforcement of smoking bans in closed public spaces.": Reference not provided.

7. "This is evident from the significant smuggling activity in the direction of Syria. The Regie’s willingness to satisfy this market demand is evident from their rapid increase in cigarette production in recent years." This statement seems to be more of an author's opinion rather than supported by primary literature.

Reviewers' comments:

Reviewer's Responses to Questions

**Comments to the Author**

1. If the authors have adequately addressed your comments raised in a previous round of review and you feel that this manuscript is now acceptable for publication, you may indicate that here to bypass the “Comments to the Author” section, enter your conflict of interest statement in the “Confidential to Editor” section, and submit your "Accept" recommendation.

Reviewer #2: All comments have been addressed

2. Is the manuscript technically sound, and do the data support the conclusions?

Reviewer #2: Partly

3. Has the statistical analysis been performed appropriately and rigorously?

Reviewer #2: Yes

4. Have the authors made all data underlying the findings in their manuscript fully available?

Reviewer #2: No

5. Is the manuscript presented in an intelligible fashion and written in standard English?

Reviewer #2: Yes

6. Review Comments to the Author

Reviewer #2: 

1. Why does the study have strengths and limitations captured together in the same section? Isn’t the strengths part of the contributions of the study? I suggest that the strengths should be captured in the contribution and the section on limitation and future research direction must stand alone, as a component of the conclusion

2. What is the limitation of the lack big tobacco firm participation on the results of the study? This must come out well

3. The manuscript needs some proof reading, some parts do not read well.

7. PLOS authors have the option to publish the peer review history of their article (what does this mean? ). If published, this will include your full peer review and any attached files.

**Do you want your identity to be public for this peer review?** For information about this choice, including consent withdrawal, please see our Privacy Policy .

Reviewer #2: No

---

## [Author Response · Author response to Decision Letter 2]

6 Feb 2025

Responses to the Editor’s and Reviewer’s Comments for Manuscript PONE-D-24-14365

A Political Economy of the Tobacco Supply Chain in an Eastern Mediterranean Country: The Case of Lebanon

By Ali Abboud, Ali Chalak, Joanne Haddad, Mariam Radwan

Dear Professor Fabrizio Ferretti,

We sincerely appreciate your handling of our submission at PLOS ONE and thank you for the opportunity to revise our manuscript.

After carefully considering the feedback, we have prepared a fully responsive revision. We agreed with all the comments and have amended the paper accordingly. We hope that the revised version meets the editorial requirements and adequately addresses the concerns of both the academic editor and reviewers. Once again, we believe this process has significantly improved the substance and presentation of our analysis.

Throughout the document, we use boxes to highlight the reviewer comments, with our responses in plain text. Unless stated otherwise, all page, table, and figure references correspond to the revised version of the manuscript.

Unlike the first round of revisions, we have not included a summary of the major changes. Instead, we directly provide a detailed, point-by-point response to the reviewers' suggestions.

Ali Abboud, Ali Chalak, Joanne Haddad, Mariam Radwan

Journal Requirements

Response: We have carefully reviewed our reference list to ensure its completeness and accuracy. To verify that none of the cited papers have been retracted, we cross-checked them against the Retraction Watch database. Based on our review, we confirm that none of the references in our manuscript have been retracted.

Additional Editor Comments

Response: We appreciate this feedback and have carefully reviewed the manuscript to ensure that all statements are properly supported by references or clearly attributed to the interviews. Where necessary, we have added citations to relevant primary literature to substantiate the claims. Additionally, we have explicitly clarified instances where information was derived from the interviews to ensure transparency.

Response: We thank the editor for highlighting the need for a citation. We have statements from both the Regie representative and the Ministry of Finance (MOF) representative supporting this claim, and we have amended the manuscript accordingly to reflect this.

Response: This information was mentioned in the interview with the Regie representative when he introduced the Regie. However, it is common knowledge and is also available on the Regie’s website (https://www.regie.com.lb/Article/1/who-we-are/en). We have added a new footnote (#2) to the revised version of the manuscript to indicate this.

Response: We thank the editor for raising this concern. We have revised the text to provide the necessary support for this claim.

We now write “Based on an interview extract with a researcher involved in tobacco monopolies and taxation policies, it is estimated that 5 to 6 billion dollars were transferred to the treasury between 1990 and 2020. These numbers align with the data provided by the Regie in its financial overview, which is reported in Fig 1.”

Response: We thank the editor for this comment. We would like to clarify how we arrived at this statement. The total subsidy cost of 1.032 billion USD between 1994 and 2021 is based on data provided by the Regie in its financial overview, which is presented in Fig 1 of the manuscript. The claim that the subsidy price is 60 to 100 percent higher than the global market price was inferred from interviews with both the Regie representative and the Farmers Union representative. They noted that in 2023, the global market price for tobacco leaves was $3 per kg, while the Regie paid an average of $5 per kg, supporting the inference that the subsidy is 60 to 100 percent higher than the global price.

We have amended the manuscript to provide more clarity to the reader on how we arrived at this statement.

Response: This statement is based on an extract from an interview with the Regie representative. We have amended the manuscript to indicate so.

Response: This statement is based on an extract from an interview with an anti-tobacco civil society activist. We have amended the manuscript to indicate so.

We now write “One of the interviewees from civil society, who was involved in the effort to introduce this law, noted that more than a decade after its passage, there has been a noticeable regression in its implementation, particularly regarding the enforcement of smoking bans in closed public spaces. Nonetheless, the interviewee noted that, despite this setback, there are some lasting effects of Law No. 174, most notably the ban on advertisements.”

Response: Indeed, you are right, this is based on suggestive evidence. Several interviewees mentioned an increase in tobacco smuggling to Syria. The Regie representative did not deny this but stated something along the lines of, “We produce, and maybe some of it goes to Syria, but we cannot control that.”

We now modify the manuscript to adjust the statement to make it less conclusive and more suggestive.

We now write “A second source of this contradiction stems from the Regie de facto operating as a supplier of tobacco (mostly cigarettes) to a larger regional market, not just the Lebanese market. This is particularly evident given indications of significant smuggling activity toward Syria and the Regie’s increased cigarette production in recent years, which may suggest a response to this market demand.”

Authors’ Response to Reviewer #2

Response: We thank the reviewer for this suggestion. We have revised the conclusion section to separate the strengths and limitations, rather than having them combined in the same section. Specifically, we now highlight both the theoretical and practical contributions of the study, in addition to its strengths. We have also created a standalone subsection dedicated to the study’s limitations and future research directions. We hope that our revisions adequately address the comments we received from you on this matter in both rounds of revisions.

All modifications have been made in track changes mode.

Response: We thank the reviewer for raising this concern. We have amended the sub-section on the limitations of the study to further discuss the limitation arising from the lack of participation from big tobacco firms. The subsection now reads as follows:

“The study has several limitations, including potential challenges in generalizing its findings to countries with significantly different supply chain structures in the tobacco industry. Additionally, relying solely on qualitative data may limit the depth of quantitative analysis needed to fully assess the economic impacts of proposed policy changes. Another limitation is the lack of participation from major tobacco firms in the study. On one hand, this is a limitation in terms of the data, as it leaves one side of the picture unexplored. On the other hand, given that this study focuses on the political economy of tobacco, the refusal of big tobacco firms to participate in the interviews may suggest a reluctance to be transparent, possibly indicating an interest in withholding information about the extent of their benefits from the current status quo.

There is also a possibility of bias in the data collected, given the strong political ties of the tobacco industry in Lebanon which may have influenced the responses of certain stakeholders, potentially introducing bias. Moreover, the feasibility and practicality of implementing the suggested policy interventions, especially within Lebanon's complex political and economic landscape, may pose significant hurdles that need careful consideration.”

Response: Thank you for your comment. We appreciate your feedback and have carefully proofread the manuscript to improve clarity and readability. We have made revisions to ensure that all sections flow smoothly and are easier to follow. Please let us know if there are any specific areas that you would like us to address further.

---

## [Editor Report · Decision Letter 2]

13 Feb 2025

A Political Economy of the Tobacco Supply Chain in an Eastern Mediterranean Country: The Case of Lebanon

PONE-D-24-14365R2

Dear Dr. Chalak,

We’re pleased to inform you that your manuscript has been judged scientifically suitable for publication and will be formally accepted for publication once it meets all outstanding technical requirements.

Kind regards,

Fabrizio Ferretti, Ph.D.

Academic Editor

PLOS ONE
---

## [Editor Report · Acceptance letter]

PONE-D-24-14365R2

PLOS ONE

Dear Dr. Chalak,

I'm pleased to inform you that your manuscript has been deemed suitable for publication in PLOS ONE. Congratulations! Your manuscript is now being handed over to our production team.

Kind regards,

on behalf of

Dr. Fabrizio Ferretti

Academic Editor

PLOS ONE